



# Dependence Models for Multi-Hazard-Events

Georg Ch. Pflug[1,2,*], Viktoria Kittler[1,*], and Stefan Hochrainer-Stigler[2]

[1]University of Vienna, Faculty of Econmics and Statistics, Oskar Morgenstern Platz 1, A- 1090 Vienna, Austria
[2]IIASA - International Institute for Applied Systems Analysis, Schlossplatz 1, A-2361 Laxenburg, Austria

**Correspondence:** Georg Ch. Pflug (georg.pflug@univie.ac.at)

**Abstract.** In recent years, the focus of research about natural hazards has turned from single-hazard studies to multi-hazard ones. While single hazards (like earthquakes, floods, droughts, etc.) have been extensively studied in the past and many quantitative statements about intensities and severities are available, quantitative studies about multi-hazards and dependencies are still rare. This paper introduces new statistical models for the dependencies of cat-event processes of different hazard types

based on Poisson-type event processes. Moreover, the models are applied to data for several natural hazard events from the Danube area in Europe. The analysis should help to bridge the gap between the more conceptual contributions to this discussion by providing empirical evidence on interactions on a large-scale region.

*Key words and Phrases*: multi-hazard models, dependent point processes, triggering of events, Danube region.

## 1 Introduction

Despite global efforts to lessen natural hazard induced disasters, such events are still on the rise, causing large direct as well as indirect losses across the globe (EM-DAT 2023). Especially climate- and weather related disasters show a continuous increase. Reason for this increase in losses can be partly attributed to increases in exposure within hazard prone areas as well as to climate change effects (IPCC 2022). While such single events caused huge impacts, multi-hazard events as well as corresponding multi-risks are gaining increasing attention due to their combined and often profound effect across various stakeholders and scales

(Ward et al. 2022). Indeed, the number of countries which are experiencing overlapping disasters (i.e. instances when two disasters happening in the same country and the second one started before or shortly after the first one ended) has increased over time (IFRC 2022). Worryingly, there are indications that such multi-hazard events have more severe impacts than single events as such situations also drain significant resources at the same time and influence risk bearers in their ability to respond (Raymound et al. 2020).

As a consequence, in recent years scientists have started to consider multiple natural hazards and their interrelationship. Studies for single hazards (like earthquakes, floods, droughts, etc.) have been extensively performed in the past and lots of quantitative statements about intensities and severities are available. In contrast, there is still a gap for tested evidence of the quantitative relationships between hazards of different type (multi-hazards) (Claassen et al. 2023).

The main question in multi-hazard research is to identify qualitative and quantitative relationships between different types

of hazards. To be more precise, suppose that for a hazard of type Y we have recorded



|      | EQ | LA | FL | DR | WF |
|------|----|----|----|----|----|
| EQ   | 3  | 3  | 2  | 0  | 0  |
| LA   | 0  | 2  | 2  | 0  | 0  |
| FL   | 0  | 2  | 2  | 0  | 0  |
| DR   | 0  | 0  | 0  | 0  | 1  |
| WF   | 0  | 1  | 1  | 0  | 2  |

**Table 1.** An Example of a quantification of dependencies

- The times of the events $T_1^Y, T_2^Y, \ldots$

- The severities of the events $X_1^Y, X_2^Y, \ldots$

- The locations of the events $L_1^Y, L_2^Y, \ldots$ (e.g. in Gauss-Krüger coordinates)

Here $Y$ may stand for a certain hazard like EQ (earthquakes), LA (landslides), FL (floods), DR (droughts), WF (wildfires),
ET (extreme temperatures), ST (storms) and so on. The units for severities depend on the type of event, e.g. peak ground acceleration (for earthquakes), flooded area or water level (for floods), duration of days (for droughts), burnt area (for wildfires), etc. It is quite obvious that some of the event types may trigger events of the other type, like earthquakes may trigger landslides and storms may trigger floods. In order to get an insight, which causal relationships are possible, Gill and Malamud (2014) presented a matrix of causalities between different types of hazards. They consider 21 types of hazards and fill a 21 by 21 table with indications of the dependency or independence. Note, this matrix is not symmetric. For our purposes we focus on a submatrix of their matrix.

The rows symbolize the triggering hazard $Y$ and the columns symbolize the triggered hazard $Z$. 0 means no causality, 1 to 3 means increasing evidence for possible causality. In order to get to such a table, Gill and Malamud Gill and Malamud (2014) introduce the notions of *spatial overlap* and *temporal likelihood*. For determining the spatial overlap, they find (by visual inspection of disaster maps) the answer of the question "In all the locations where the primary hazard is present, what proportion of these could occurrences of the secondary hazard also occur?". The temporal likelihood is found by a "qualitative analysis of reviewed literature, which enabled an understanding of the relative occurrence of secondary hazards after primary cases of a primary hazard." A point score for spatial overlap (1=limited 2=medium, 3=large) as well as for the temporal likelihood (1=low, 2=medium, 3=high) was defined. The product of the two values gives the overlap-likelihood factor, which can be defined for each triggering $\Rightarrow$ triggered event relationship. In addition, "a more mechanistic approach, using a form of engineering judgement" complemented this review of case studies." According to the two judgements, a score for the degree of dependency (in our above example 0,1,2,3) was assessed. They call the highest degree of dependency as "triggering" and a lower degree as "increasing probability".




Contrary to this approach we do not distinguish in our paper between "triggering a secondary event " and "increasing the probability of a secondary event" as Gill and Malamud do. Rather we argue that it is necessary to distinguish whether the primary event influences the intensity (the timing) of the secondary event or the severity (the magnitude) of the secondary event or both. Under the assumption that the primary event influences timing of the secondary event, by changing its intensity from $\lambda_0$ to $\lambda_1$, one may set $\lambda_1$ to infinity to model immediate triggering. Values of $\lambda_1$ larger than $\lambda_0$, but less than infinity

model the "increasing probability" of the secondary event. We introduce a set of observations from the Danube Region and two discrete event processes to estimate and test the dependency by statistical methods. In doing so, the expert opinion of a relationship matrix of the above type is an important guideline to select the relationships one wants to test in a second step with empirical data.

     We argue that qualitative statements about multi-hazard relationships need a quantitative argument based on data observation

to become a statistically significant relationship. Our contribution to the discussion of multi-hazard events is in the introduction of some statistical models to prove such relationships and give concrete examples using real data. To start with our discussion, the next section introduces two models for estimating and testing the triggering effect (i.e. the occurrence of event type $Y$ triggers the occurrence of event $Z$, in symbol: $Y \Rightarrow Z$) and afterwards introduces the case study and data used and the results found. Finally, we conclude with a discussion and outlook to the future.

## 2   Methodology

Before introducing the suggested models, some overview over the well known existing models of Poisson-type event processes is given. Since the event times are ordered $T_1 < T_2 < T_3 < \ldots$, the inter-event times $V_1 = T_2 - T_1, V_2 = T_3 - T_2, \ldots$ are non-negative random variables. If they are mutually independent, we speak of a *renewal process*. Let $N(t_1, t_2)$ be the (random) number of events in the interval $[t_1, t_2]$. For any discrete event process, we may formulate the *intensity* $\lambda_t$ as the instantaneous event rate:

$$\lambda_t = \lim_{h \to 0} \frac{1}{h} \mathbb{P}\{N(t, t+h) > 0 \mid \text{there is an event at time t}\}.$$

Then the expected number of events in $[t_1, t_2]$ is

$$\mathbb{E}[N(t_1, t_2)] = \int_{t_1}^{t_2} \lambda_t \, dt$$

The most important special case is the *homogeneous Poisson process* which has constant intensity $\lambda$, implying that the $V_t$'s are i.i.d. Exponential variables with expectation $\frac{1}{\lambda}$. For a homogeneous Poisson process with constant intensity $\lambda$ we have that $N(t_1, t_2)$ follows a Poisson distribution with expectation

$$\mathbb{E}[N(t_1, t_2)] = (t_2 - t_1)\lambda$$





If the intensity is time-dependent, the process is an *inhomogeneous Poisson process*. For an inhomogeneous Poisson process with intensity function $\lambda_t$, the next event time $T_{i+1}$ after an event at time $T_i = t$ follows the conditional distribution function

$$\mathbb{P}(T_{i+1} \leq t + u \mid T_i = t) = 1 - \exp\left(\int_t^{t+u} \lambda_s \, ds\right).$$

If the severities $X_t$ are included in the model, we speak of a marked Poisson process. Here, one assumes that the severities $X_t$ are i.i.d. and independent of the inter-event times $V_t$. However, for some types of events, there may be a dependency between $V_t$ and $X_t$ expressed by some copula[1]. For instance, a model with damage sizes $X_t$ depending on the inter-arrival times $D_t$ in insurance mathematics was developed in Pflug and Mueller (2001).

### 2.1 Inter-dependencies of event processes of different types

Consider now two types of hazards $Y$ and $Z$ and the pertaining event time processes $T_1^Y, T_2^Y, \ldots$ respectively $T_1^Z, T_2^Z, \ldots$, each following a homogeneous Poisson process with intensities $\lambda^Y$ and $\lambda^Z$ respectively. We call $Y$ the primary (triggering) hazard and $Z$ the secondary (triggered) hazard. One of the first models is the self-exciting Poisson model by Hawkes (Hawkes (1971a), Hawkes (1971b(@))). It assumes that past events of type $Y$ influence events of type $Z$ and vice-versa. It reads

$$\lambda_t^Y = v^Y + \sum_{T_i^Y \leq t} \gamma^Y(t - T_i^Y) + \sum_{T_j^Z \leq t} \gamma^{YZ}(t - T_j^Z)$$

$$\lambda_t^Z = v^Z + \sum_{T_j^Z \leq t} \gamma^Z(t - T_j^Z) + \sum_{T_i^Y \leq t} \gamma^{ZY}(t - T_i^Y)$$

where $\gamma^Y, \gamma^Z, \gamma^{YZ}, \gamma^{ZY}$ are *forgetting functions*, which determine the influence of past events onto the future intensity of events of same type or of the other type. Hawkes indicates a method to calculate the covariance of the intensities based on the forgetting functions. A special case is the univariate model

$$\lambda_t^Y = v^Y + \sum_{T_i^Y \leq t} \gamma^Y(t - T_i^Y)$$

which is known as *self-exciting point process*, since the occurrence of type $Y$ triggers a change of intensity of the same process. Such a model can be used for contagious processes, like infections, but not used here, since the focus in this study is about the relation of different natural hazards.

Another model of this type is due to Doss (1989), going back to an idea of Ripley (1977)). Let as before $N^Y(t_1, t_2)$ be the number of events of type $Y$ with expectation $\lambda^Y(t_2 - t_1)$. In case of independence, the expression

$$K(t_1, t_2) = \frac{1}{\lambda^Y} \, \mathbb{E}[N^Y(t_1, t_2) \mid \text{an event of type Z occurred at time 0}]^2$$

---

[1]A copula is a function that splits multivariate distribution functions into two parts. The name derives from the word 'to couple'. The univariate marginal distributions are separated from the copula function, which describes the dependency structure of the random variables Nelson (2006).

[2]It can be shown that this expression is symmetric in $Y$ and $Z$ in the following way:
$K(t_1, t_2) = \frac{1}{\lambda^Z} \, \mathbb{E}[N^Z(t_1, t_2) \mid \text{an event of type Y occurred at time 0}]$


equals $t_2 - t_1$. To test this for data in the interval $[0, T]$ with $n^Y$ $Y$-observations and $n^Z$ $Z$-observations one may use the test statistic

$$\hat{K}(t_1, t_2) = \frac{T}{n^Y n^Z} \sum_{i=1}^{n^Y} \sum_{j=1}^{n^Z} \mathbb{1}_{T_i^Y - T_j^Z \in (t_1, t_2)}$$

where $\mathbb{1}$ is the indicator function. Doss shows some properties of this test. Note, both models, Hawkes and Doss are symmetric, that is the roles of $Y$ and $Z$ are exchangeable. This is however not always justified, since their is often a clear causality between types of hazards: An earthquake may cause a landslide, but a landslide will not cause an earthquake. For this reason, asymmetric models seem to be more appropriate. In the next section we therefore introduce triggering models.

## 2.2   Triggering models

Assume now that we observe two processes $Y$ and $Z$ and ask whether we may prove or disprove the triggering effect $Y \Rightarrow Z$. We will discuss two related models next, the switching intensity model as well as the first follower times model.

### 2.2.1   The switching intensity model

If process $Y$ has an event at time $t$, it may increase or decrease the intensity of process $Z$ for some time. Suppose that the stationary intensity of process $Z$ is $\lambda_0^Z$. Then one may consider the following model

$\lambda^Z(t) = \begin{cases} \lambda_0^Z & \text{if no event of process Y happened in the period } [t - \delta, t] \\ \lambda_0^Z + a & \text{if at least one event of process Y happened in the period } [t - \delta, t] \end{cases}$

Here $a$ denotes the shift in intensity caused by process $Y$ and $\delta$ denotes the duration of the effect of $Y$ on $Z$. These parameters can be easily estimated from data.

An alternative model would assume a decreasing effect of $Y$ on the intensity of $Z$:

$$\lambda^Z(t) = \lambda_0^Z + a(t - T^Y) \quad \text{where } T^Y \text{ is the last event of type Y before } t$$

Here $a(t)$ is a forgetting function like

$a(t) = a\mathbb{1}_{t \in [0, \delta]}$ (the previous example), or

$a(t) = a \exp(-\alpha(t))$ (the exponential forgetting)

An example for the statistical estimation of such models for the relation DR $\Rightarrow$ WF, answering the question how much does the occurrence of drought events trigger wildfires is contained in Section 4.1.

### 2.2.2   The first follower times model

If an event of type $Y$ happens at time $t$, then denote by $\tau^Z(t)$ the (random) time of the next event of process $Z$. The difference

$$W_t^Z = \tau^Z(t) - t$$


is called the *first follower time*. Under the independence hypothesis, the distribution of the first follower times $W_t^Z$ equals
the distribution of the inter-event times $V_t^Z$ of process $Z$. Any deviation form this equality can be interpreted as dependency
of process $Z$ on process $Y$. We may test the dependency by performing a Kolmogorov-Smirnov test for the equality of the
distributions $W_t^Z$ and $V_t^Z$. Also this model was used to test the triggering effect DR $\Rightarrow$ WF, see Section 4.1.

## 2.3    Dependent locations

If $X_t^{Y,1}$ and $X_t^{Y,2}$ are processes of the same type, but at different locations occurring at the same or nearly same event times,
then these severity variables may be dependent (and their degree of dependency may depend on the distance of the two
locations). We may estimate the degree of dependency by a copula between them as was done, for example, for flood risk on
the Pan-European level Timonina et al. (2015).

The location model can be incorporated into the timing model: Suppose that the $Y$-process has an event at time $t_0$ and
location $L_t^Y$. Then one may consider the model

$$\lambda_t^Z = \lambda_t^Z\left(t - t_0, dist(L_t^Z, L_{t_0}^Y)\right)$$

where $dist$ is the distance between the locations, such that the intensity jump becomes smaller for more distant locations. Due
to the lack of exact data, this model was not implemented for this paper.
The previous other two models were however applied for our case study as it is described next.

## 3    Case Study: Region and Data Used for Testing DR $\Rightarrow$ WF

We selected the Danube Region as our case study as the area is exposed to various disaster types that also caused huge losses
in the past. It is also very heterogeneous in economic terms and it therefore can be expected that single and multi-hazard events
will have quite some significant impact across countries as well as within countries. The entire data set from EMDAT consisted
of 2918 observations and 50 variables. In a first step some selections have been made, like the restriction on the Danube region
and in our analysis only wildfires and droughts are taken into account in section 4. Afterwards, in section 5 some additional
analysis for other hazards as given in Table 1 are looked at. In addition, the countries in the data set had to be examined
more closely, since some of it consisted of outdated country names, such as the Soviet Union, Yugoslavia, Czechoslovakia,
Serbia-Montenegro, as well as the German Democratic Republic and German Federal Republic.
Specifically, the following countries with the corresponding ISO 3166-ALPHA3 code are considered (in the Appendix
one can also find a map of Europe with the countries to be considered, i.e. the Danube region): Austria (AUT), Bosnia and
Herzegovina (BIH), Bulgaria (BGR), Croatia (HRV), Czech Republic (CZE), Germany (DEU), Hungary (HUN), Moldova
(MDA), Montenegro (MNE), Romania (ROU), Serbia (SRB), Slovakia (SVK), Slovenia (SVN) and Ukraine (UKR).
Since the duration of the respective events will play a major role for further analysis, the next step was to compute a date
variable (specifically Start Month, Start Day and End Month, End Day). This is done by 'k-nearest-neighbour imputation'.
After this, the duration was determined as the difference between the end and start dates. The final resulting data set now



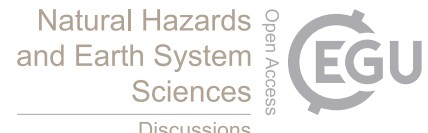

| Country | Drought | Earthquake | Extreme temp. | Flood | Landslide | Storm | Wildfire |
|---|---|---|---|---|---|---|---|
| | DR | EQ | ET | FL | LS | ST | WF |
| AUT | 0 | 1 | 6 | 18 | 5 | 20 | 0 |
| BIH | 2 | 2 | 4 | 17 | 1 | 2 | 0 |
| BGR | 2 | 5 | 9 | 21 | 0 | 6 | 5 |
| HRV | 1 | 3 | 6 | 12 | 0 | 2 | 8 |
| CZE | 0 | 0 | 9 | 14 | 0 | 15 | 0 |
| DEU | 0 | 3 | 14 | 22 | 1 | 59 | 1 |
| HUN | 3 | 1 | 6 | 16 | 0 | 8 | 0 |
| MDA | 3 | 2 | 3 | 7 | 0 | 2 | 0 |
| MNE | 0 | 2 | 2 | 8 | 0 | 0 | 2 |
| ROU | 2 | 4 | 20 | 52 | 1 | 10 | 0 |
| SRB | 1 | 3 | 11 | 24 | 0 | 1 | 0 |
| SVK | 0 | 0 | 5 | 15 | 2 | 3 | 1 |
| SVN | 0 | 2 | 2 | 5 | 0 | 3 | 0 |
| UKR | 1 | 0 | 8 | 18 | 0 | 10 | 2 |
| | 15 | 28 | 105 | 249 | 10 | 141 | 19 |

**Table 2.** The number of events recorded in the coutries of the (extended) Danube region

consisted of 567 observations. The absolute and relative frequencies of each disaster in the form of a pie chart can be found in the Appendix. Summarizing this chart, the majority of events are weather and climate related, with 44 percent being drought and 25 percent being storm as well as 19 percent extreme temperature. Moreover, one can analyze the occurrence of each disaster type for a certain country. This can be seen in the following table.

The table indicates that some countries experience many hazards while some others relatively few ones. In addition some of them are happening rather frequently in some countries while others only happened in a few instances. In section 4 only the disasters 'wildfire' and 'drought' are analyzed and this table can already be used to determine minor indicators for their joint occurrence. For instance, in Bulgaria were two droughts and five wildfires recorded. Nevertheless, the possible dependency with the previously mentioned models (see section 2.2) will be statistically verified in the following chapter. Afterwards, section 5 presents some additional hazard interaction are analyzed based on Table 1.

## 4   Analysis and Results

In this chapter the above mentioned models are applied to the data from the Danube Region. In general, wildfires are caused by a mixture of factors such as high temperatures, drought conditions following a period of vegetation growth and a trigger which

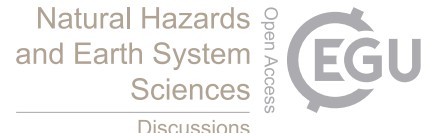

| Drought | | Wildfire | |
|---|---|---|---|
| Date | Location | Date | Location |
| 1983-05-18 | BGR, ROU | 1983-07-25 | DEU, HRV |
| 1986-06-14 | HUN | 1985-08-06 | HRV |
| 1990-07-13 | SRB | 1997-07-29 | HRV |
| 1992-05-24 | HUN | 2000-07-01 | BGR |
| 2000-03-08 | BGR | 2000-08-10 | HRV, MNE |
| 2000-05-11 | MDA | 2000-10-26 | SVK |
| 2000-06-19 | ROU | 2001-08-10 | BGR |
| 2000-08-10 | BIH | 2003-07-18 | HRV |
| 2003-02-20 | HRV | 2007-07-15 | BGR |
| 2003-05-16 | BIH | 2007-08-04 | HRV |
| 2003-07-18 | HUN | 2007-08-25 | BGR |
| 2007-07-15 | MDA | 2007-08-30 | HRV |
| 2012-04-15 | UKR | 2017-07-16 | HRV, MNE |
| 2012-11-05 | MDA | 2020-07-06 | UKR |
| | | 2020-09-30 | UKR |
| | | 2021-08-02 | BGR |

**Table 3.** The timing and the location of the events used in our analysis

can be natural such as lightning[3]. This statement will now be examined with the help of the available data. Table 3 shows the event (starting-) times for both hazards and their locations.

As can be seen here, there are only single events for a certain country. Therefore, it makes little sense to look at the data separately for each country, since at most two points in time can match, otherwise the time span would be too long. The dates

of both hazards, when an event took place, are illustrated in Figure 1.

In Figure 1 one can see that the intensities for both hazards have increased over time. The increase in wildfires can (graphically) be explained by previous droughts or other influences. Droughts may have become more frequent due to climate change and the extreme temperatures it causes.

The following four graphs represent the number of events per year as well as the cumulative sum of events for both hazards

(separately for drought and wildfire).

---

[3]https://www.n-d-a.org/fire.php


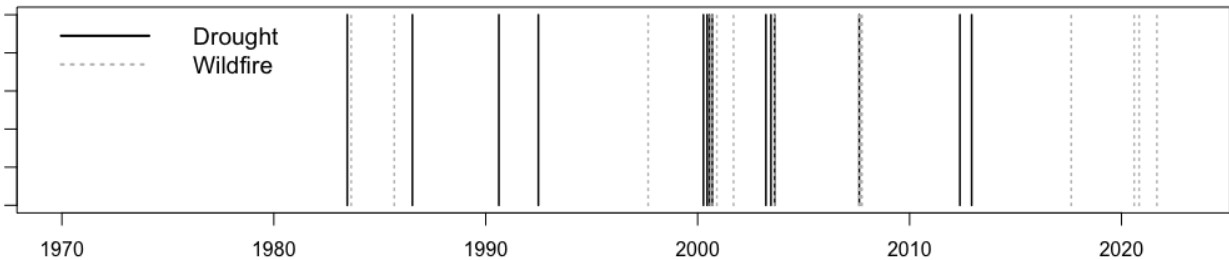

**Figure 1.** Event-times of drought and wildfire (total observations)

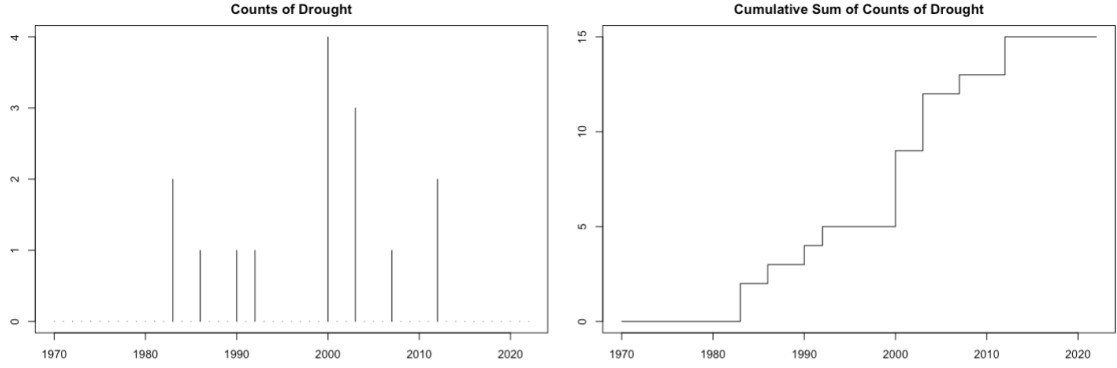

**Figure 2.** Counts of Drought (left), Cumulative Sum of Counts of Droughts (right)

In the following three naive attempts to estimate the *'Intensity function'* are graphically presented, where in general the intensity is estimated by $\lambda_t = \frac{\text{number of events}}{\text{time interval in years}}$.

The first graph, Figure 4, shows the naive estimated intensity function for all countries together which separates each year into an interval. That is, for each year $t$ the intensity is defined as

$$\lambda_t = \{\#T_i : t \le T_i < t+1\}.$$

Another way of estimating the intensity can be done by creating intervals between each pair of events occurring. Then the intensity will be calculated as follows:

$$\lambda_t = \frac{1}{T_{i+1} - T_i} \qquad \text{for } T_i \le t < T_{i+1}.$$





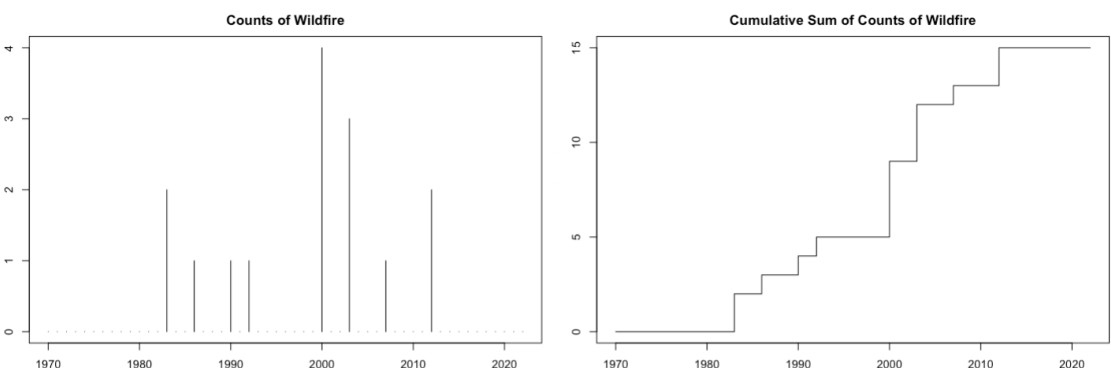

**Figure 3.** Counts of Wildfires (left), Cumulative Sum of Counts of Wildfires (right)

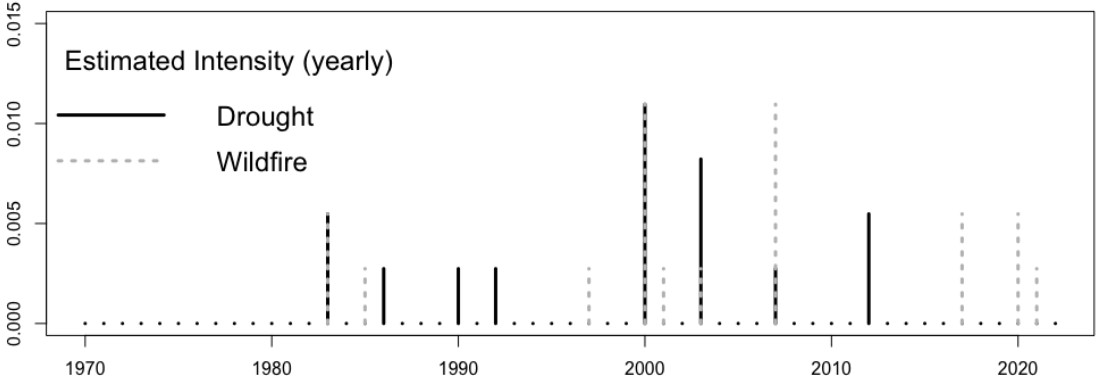

**Figure 4.** Estimated intensity function for each year

Here it is possible to distinguish between two methods. On the one hand the inclusion of the first event $([1970, T_1], (T_1, T_2], \ldots, (T_n, 2022]$
or on the other hand the exclusion $([1970, T_1), [T_1, T_2), \ldots, [T_n, 2022])$. In figure 5 the upper picture shows that the first hit will

be included and in the lower picture excluded.

One could make the attempt to model droughts or wildfires without taking the (possible) influence of another hazards into account. Computing the intensity function via Maximum Likelihood gives the following estimates:

These estimates lead to the graphical representation in Figure 6. Here one can see that the intensity function of wildfires increases over time while for droughts the intensity remains constant.

In the next part, we want to analyze if the occurrence of wildfires is triggered by droughts or not.



Natural Hazards
and Earth System
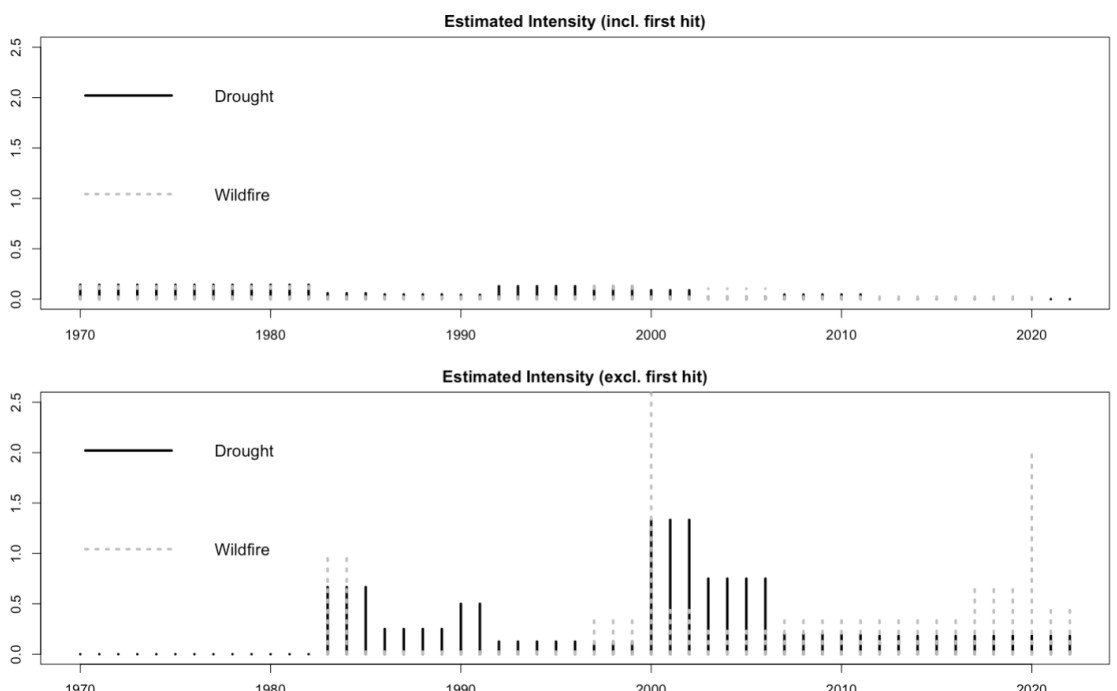

**Figure 5.** Estimated Intensity function incl. and excl. first hit

| (Drought) | Estimate | STD. Error |
|---|---|---|
| Intercept | -6.176 | 0.561 |
| Time | 0.012 | 0.017 |
| (Wildfire) | Estimate | STD. Error |
| Intercept | -6.845 | 0.629 |
| Time | 0.040 | 0.016 |

**Table 4.** Testing the time-dependency of WF on DR

## 4.1 Triggering

An obvious reason for the occurrence of a wildfire is a preceding period of drought. Consequently, we want to try to determine a connection between the two events. In terms of the *switching intensity model*, a dummy variable is created that indicates whether a drought phase was documented in the respective country and the surrounding countries in the previous 3 months.

Analysis of droughts separately for Bulgaria showed that there was only one drought up to 3 months before. Namely in Romania on June 19th, 2000. For the analysis of wildfires separated for Croatia the number of droughts in the surrounding countries gave the following: A wildfire was documented on August 8th, 2000, and in Bosnia and Herzegovina a drought





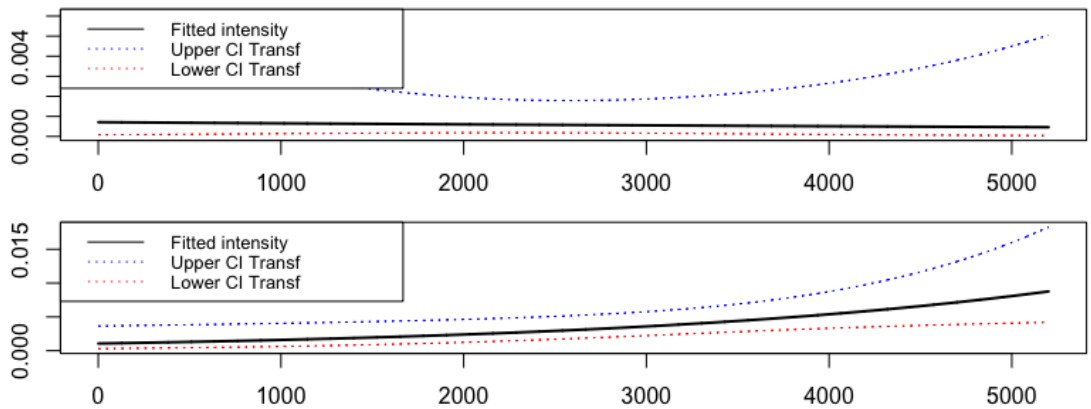

**Figure 6.** Intensity function Maximum Likelihood (DR above, WF below)

was recorded previously. Equally, a wildfire was documented on July 7th, 2003, and in Croatia, Bosnia and Herzegovina and Hungary droughts were observed previously. For the analysis of wildfires in total, the number of droughts preceded by up to 3

170   months is as follows: Bulgaria (1), Romania (2), Bosnia and Herzegovina (1), Croatia (1), Hungary (1) and Moldova (1). As mentioned before, we model all observations together due to the rare events.

### 4.1.1   The switching intensity model

In Section 2.2.1 the switching intensity model was introduced as

$$
\lambda^Z(t) = \begin{cases} \lambda_0^Z & \text{if no event of process Y happend in the period } [t - \delta, t] \\ \lambda_0^Z + a & \text{if at least one event of process Y happend in the period } [t - \delta, t] \end{cases}
$$

To estimate the intensity function one can use naive versions like before or calculate it via maximum likelihood, as will be done further on.

Taking all countries into account, the results for estimating a Poisson process for wildfire would give the following estimation.

Model validation via AIC shows that the third model, which includes time and previously droughts as covariates, is the best

because of the smallest AIC value. According to the likelihood ratio test below, model 3 is again the best choice. Including both covariates into the model significantly improves the fit, as reflected in the p-value. Even if only the dummy variable for previously droughts is included in the model, a significant result is obtained.

The following pictures show the estimated intensity function via Maximum Likelihood. The first graph illustrates model 1, where only the dummy variable for previously droughts is used. The second graph shows model 2, where only time is used as

covariate. And in the third graph model 3 can be seen, which includes both covariates.





|  |  | Estimate | STD. Error | AIC |
|---|---|---|---|---|
| **MOD1** | Intercept | -5.902 | 0.267 | 234.86 |
| (Dummy-DR) | Dummy | 3.142 | 0.520 | |
| | | | | |
| **MOD2** | Intercept | -6.845 | 0.629 | 248.91 |
| (Time) | Time | 0.040 | 0.016 | |
| | | | | |
| **MOD3** | Intercept | -7.233 | 0.743 | 231.45 |
| | Time | 0.043 | 0.019 | |
| (Dummy-DR +Time) | Dummy | 3.045 | 0.522 | |

**Table 5.** Testing 3 models for the intensity of WF depending on DR

| Model 1: mod3 | | | | | | Model 1: mod3 | | | | | |
|---|---|---|---|---|---|---|---|---|---|---|---|
| Model 2: mod2 | | | | | | Model 2: mod1 | | | | | |
| | #Df | LogLik | Df | Chisq | Pr(>Chisq) | | #Df | LogLik | Df | Chisq | Pr(>Chisq) |
| 1 | 3 | -112.73 | | | | 1 | 3 | -112.73 | | | |
| 2 | 2 | -122.46 | -1 | 19463 | 1.026e-05 *** | 2 | 2 | -115.43 | -1 | 5.409 | 0.02003 * |

**Table 6.** Likelihood Ratio tests for the 3 models

### 4.1.2 The first follower model

As already mentioned in section 2.2.2, the first follower times are calculated as follows.

$$W_t^Z = \tau^Z(t) - t$$

where $t$ denotes the time when an event of type $Y$ (= Drought) happens and $\tau^Z(t)$ is the next event of process Z (= Wildfire). The "inter-event times" of process Z are calculated via this rule

$$V_t^Z = T_{t+1}^Z - T_t^Z$$

where $T^Z$ denotes the time when an event of type Z (= Wildfire) occurred.

Concretely for this data, after the dates were transformed into a numerical values (for example 1983-05-18 was coded as 1983.467).we got the row data as shown in Table 7.

If the distribution of the first follower times $W_t^Z$ equals the distribution of the inter-event times $V_t^Z$ of process $Z$, then these two processes are mutually independent. To test the dependency a Wilcoxon rank-sum test was performed to test the equality of the distributions $W_t^Z$ and $V_t^Z$. Its results are presented below.

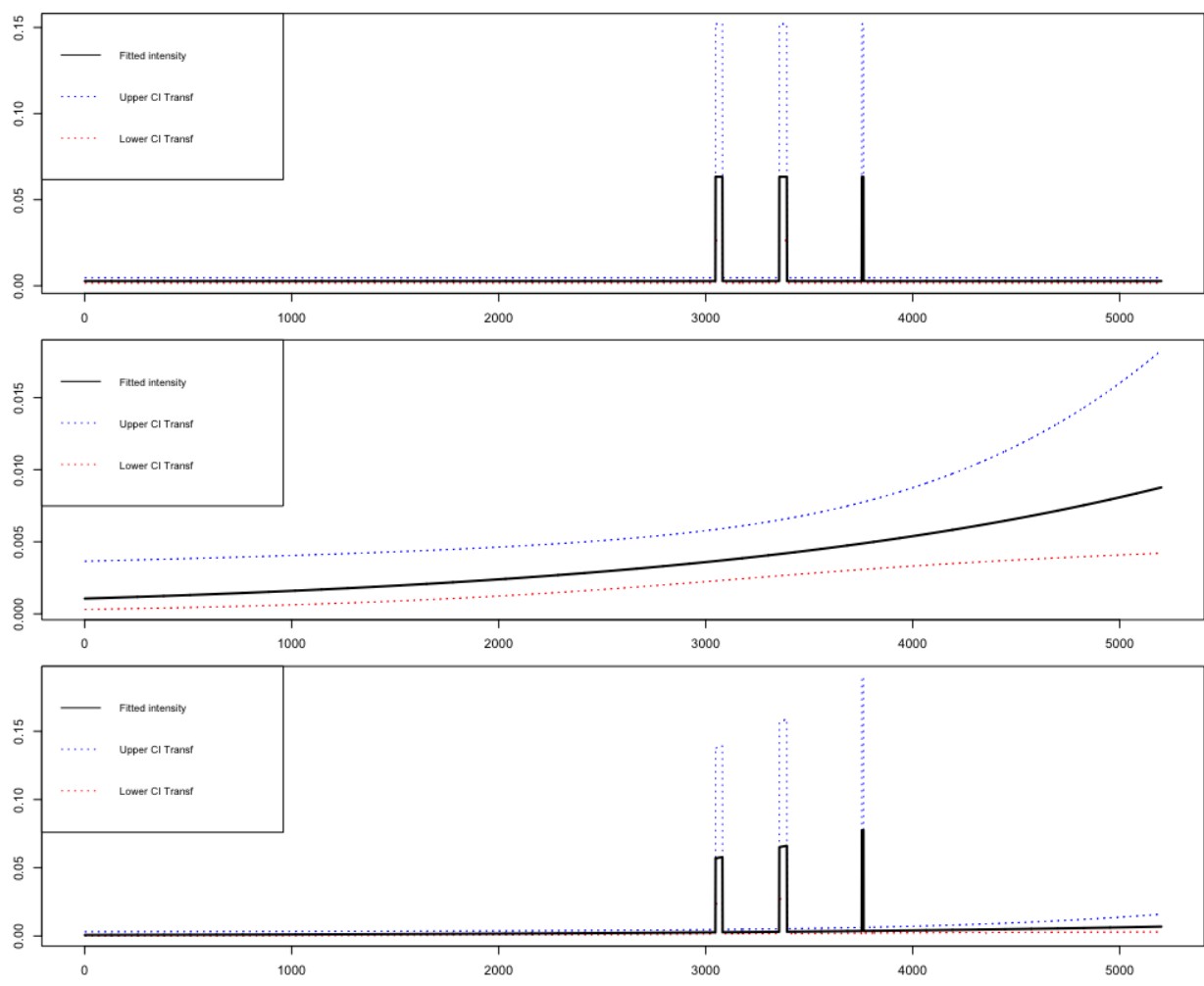

**Figure 7.** Estimated intensity function with covariates via Maximum Likelihood

According to this test, the two distributions are significantly different, that is the first follower times are significantly smaller than the inter-event times in probability, proving that the WF process depends on the DR process.

## 5 Further Results on Testing Dependencies

The aim of this paper is to introduce models for dependent Poisson processes using wildfires and droughts as explicit example. Nevertheless, the following chapter provides a brief overview of other (possible) dependent hazards.





| First Follower Time | Inter-Event Time |
|---|---|
| 0.186 | 2.030 |
| 11.125 | 11.980 |
| 7.044 | 2.922 |
| 5.180 | 0.108 |
| 0.313 | 0.211 |
| 0.138 | 0.788 |
| 0.033 | 1.938 |
| 0.000 | 3.991 |
| 0.411 | 0.052 |
| 0.172 | 0.058 |
| 0.000 | 0.013 |
| 0.000 | 9.877 |
| 5.252 | 2.972 |
| 4.697 | 0.233 |
| | 0.838 |

**Table 7.** The raw data: First follower times and inter-event times from our data set

| Two-sample Wilcoxon rank-sum test |
|---|
| T= 76, p-value = 0.0101 |
| alternative hypothesis: two-sided |

**Table 8.** Testing the dependence of the WF process on the DR process

## 5.1 Extreme Temperatures ⇒ Drought

The process of droughts is driven by the heat of the sun. The hotter it is, the greater the rate of evaporation. Thus, if the temperature of the ocean or the surface of the land is relatively cool in a certain area, drought may occur in regions that rely on those sources of moisture [4].

In terms of the switching intensity model, taking all countries into account, the results for estimating a Poisson process for drought would give the following estimation. Model validation via AIC shows that the first model, which only includes the dummy variable for previously extreme temperatures (ET) as covariates, is the best because of the smallest AIC value.

---

[4]https://kids.lovetoknow.com/learning-at-home/what-causes-drought


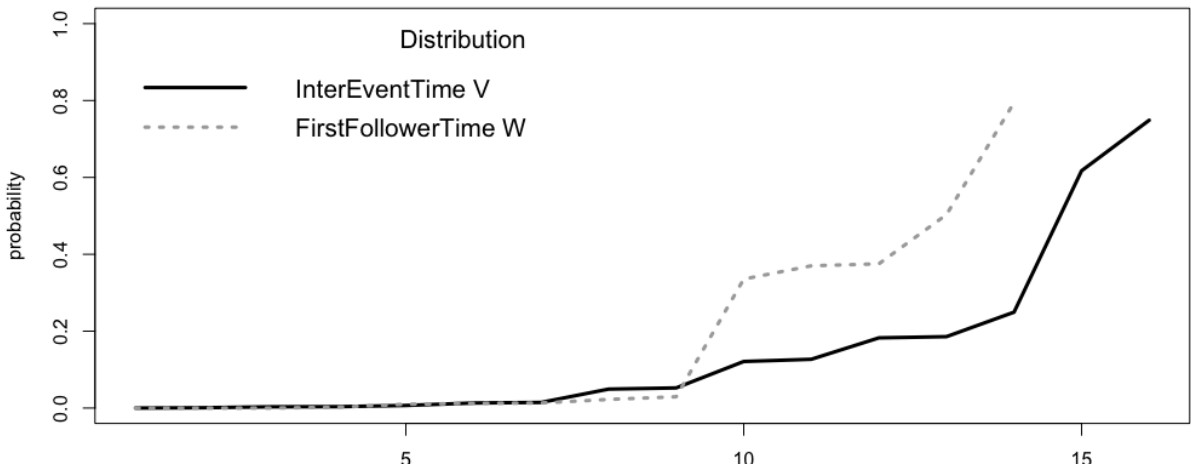

**Figure 8.** Distribution of first-follower- and inter-event-times

| | | Estimate | STD. Error | AIC |
|---|---|---|---|---|
| **MOD1** | Intercept | -6.054 | 0.289 | 199.83 |
| (Dummy-ET) | Dummy | 2.632 | 0.645 | |
| | | | | |
| **MOD2** | Intercept | -6.177 | 0.561 | 208.97 |
| (Time) | Time | 0.012 | 0.017 | |
| | | | | |
| **MOD3** | Intercept | -6.216 | 0.592 | 201.73 |
| | Time | 0.006 | 0.019 | |
| (Dummy-ET+Time) | Dummy | 2.583 | 0.659 | |

**Table 9.** Testing 3 models for the intensity of DR depending on ET

According to the likelihood ratio test below, model 3 is more suitable than model 2 because of the small p-value. However, the LR test results in that model 1 is better than model 3. That is, including both covariates into the model doesn't improve the fit significantly - only the covariate for previously extreme temperatures is significant.

Since the first model is statistically significant, it follows that a relationship between droughts and previously extreme temperatures could be proved in the switching intensity model.

In case of the first follower model, the Wilcoxon rank-sum failed to achieve a significant result. It means that the two distributions are not significantly different and the two processes are mutually independent.

In summary, it can be said that a dependency could only be determined using the switching intensity model.





| | #Df | LogLik | Df | Chisq | Pr(>Chisq) | | #Df | LogLik | Df | Chisq | Pr(>Chisq) |
|---|---|---|---|---|---|---|---|---|---|---|---|
| Model 1: mod3 | | | | | | Model 1: mod3 | | | | | |
| Model 2: mod2 | | | | | | Model 2: mod1 | | | | | |
| 1 | 3 | -97.86 | | | | 1 | 3 | -97.86 | | | |
| 2 | 2 | -102.49 | -1 | 9.247 | 0.002359 ** | 2 | 2 | -97.92 | -1 | 0.104 | 0.7468 |

**Table 10.** LR tests for the 3 models

Two-sample Wilcoxon rank-sum test

T= 315, p-value = 0.7121
alternative hypothesis: two-sided

**Table 11.** Testing the dependency of the DR process on the ET process

## 5.2 Extreme Temperatures ⇒ Wildfire

In general, wildfires are caused by a mixture of factors such as high temperatures, drought conditions following a period of vegetation growth and a trigger which can be natural such as lightning [5].

The connection between wildfires and droughts has already been shown in the previous chapter (see chapter 4). Lightning is a potential cause of a wildfire, however the analysis could not determine a connection because no observations with a suitable time period were recorded in the data. In the following the dependence of wildfires and extreme temperatures will be studied.

In terms of the switching intensity model, taking all countries into account, the results for estimating a Poisson process for wildfire would give the following estimation. Model validation via AIC shows that the third model, which includes both covariates, is the best because of the smallest AIC value.

According to the likelihood ratio test below, model 3 that is, including both covariates into the model improve the fit significantly.

Since the third model is statistically significant, it follows that a relationship between wildfires and previously extreme temperatures could be proved in the switching intensity model.

In case of the first follower model, the Wilcoxon rank-sum achieved a significant result on the 10 percent level. According to this test, the two distributions are significantly different, that is those two distributions are dependent.

In summary, it can be said that a dependency of wildfire and previously extreme temperatures could be determined using the
switching intensity model and the first follower model.

---

[5]https://www.n-d-a.org/fire.php





|  |  | Estimate | STD. Error | AIC |
|---|---|---|---|---|
| **MOD1** | Intercept | -6.144 | 0.302 | 212.98 |
| (Dummy-ET) | Dummy | 3.906 | 0.465 |  |
|  |  |  |  |  |
| **MOD2** | Intercept | -6.846 | 0.629 | 248.92 |
| (Time) | Time | 0.041 | 0.017 |  |
|  |  |  |  |  |
| **MOD3** | Intercept | -7.349 | 0.800 | 211.23 |
|  | Time | 0.039 | 0.022 |  |
| (Dummy-ET+Time) | Dummy | 3.738 | 0.465 |  |

Table 12. Testing 3 models for the WF depending on ET

| Model 1: mod3 |  |  |  |  |  | Model 1: mod3 |  |  |  |  |  |
|---|---|---|---|---|---|---|---|---|---|---|---|
| Model 2: mod2 |  |  |  |  |  | Model 2: mod1 |  |  |  |  |  |
|  | #Df | LogLik | Df | Chisq | Pr(>Chisq) |  | #Df | LogLik | Df | Chisq | Pr(>Chisq) |
| 1 | 3 | -102.61 |  |  |  | 1 | 3 | -102.61 |  |  |  |
| 2 | 2 | -122.46 | -1 | 39.69 | 2.977e-10 *** | 2 | 2 | -104.49 | -1 | 3.7534 | 0.0527 . |

Table 13. Likelihood Ratio test of WF depending on ET

## 5.3 Drought ⇒ Landslide

Landslides are caused by disturbances in the natural stability of a slope. They can accompany heavy rains or follow droughts, earthquakes, or volcanic eruptions [6].

---

[6]https://www.cdc.gov/disasters/landslides.html

Two-sample Wilcoxon rank-sum test

T= 630, p-value = 0.102
alternative hypothesis: two-sided

Table 14. Testing the dependency of the WF process on the ET process



A possible reason for the occurrence of a landslide is a preceding period of drought. Consequently, we want to try to determine a connection between the two events. Analysis of landslides in total showed that there was one drought up to 3 months before (this drought had a duration of over 350 days). Namely in Bulgaria on March 8h, 2000. Since droughts were rarely observed before landslides, a statistical connection between these two hazards cannot be concluded.

Another possible connection could exist between landslides and earthquakes. However, no observations of earthquakes before landslides were observed in all countries, and therefore a statistical relationship between these two hazards cannot be concluded.

## 5.4 Extreme Temperatures ⇒ Flood

The most common cause of flooding is water due to rain and/or snowmelt that accumulates faster than soils can absorb it, or rivers can carry it away [7].

A possible reason for the occurrence of floods may be preceding extreme temperatures. These temperatures could be the reason for snowmelt, which in turn leads to large amounts of water [8].

In terms of the switching intensity model, taking all countries into account, the results for estimating a Poisson process for flood would give the following estimation. Model validation via AIC shows that the second model, which only includes time as covariate, is the best because of the smallest AIC value.

|  |  | Estimate | STD. Error | AIC |
|---|---|---|---|---|
| **MOD1** | Intercept | -3.132 | 0.069 | 2000.48 |
| (Dummy) | Dummy | 0.695 | 0.166 | |
| | | | | |
| **MOD2** | Intercept | -4.497 | 0.184 | 1907.75 |
| (Time) | Time | 0.047 | 0.005 | |
| | | | | |
| **MOD3** | Intercept | -4.569 | 0.195 | 1908.08 |
| | Time | 0.050 | 0.006 | |
| (Dummy+Time) | Dummy | -0.236 | 0.185 | |

**Table 15.** Testing the 3 models for FL depending on ET

According to the likelihood ratio test below, model 3 is equally good as model 2, therefore model 2 is more appropriate. However, the LR test results in that model 3 is significantly better than model 1. That is, including both covariates into the model improves the fit significantly.

---

[7]https://www.weather.gov/safety/flood-hazards

[8]https://www.epa.gov/climate-indicators/climate-change-indicators-river-flooding



| | #Df | LogLik | Df | Chisq | Pr(>Chisq) | | #Df | LogLik | Df | Chisq | Pr(>Chisq) |
|---|---|---|---|---|---|---|---|---|---|---|---|
| Model 1: mod3 | | | | | | Model 1: mod3 | | | | | |
| Model 2: mod2 | | | | | | Model 2: mod1 | | | | | |
| 1 | 3 | -951.04 | | | | 1 | 3 | -951.04 | | | |
| 2 | 2 | -951.88 | -1 | 1.6665 | 0.1967 | 2 | 2 | -998.24 | -1 | 94.391 | 2.2e-16 *** |

**Table 16.** Likelihood Ratio test of FL depending on ET

Two-sample Wilcoxon rank-sum test

T= 7508, p-value = 0.007091
alternative hypothesis: two-sided

**Table 17.** Testing of the dependency of the FL process on the ET process

Since the third model is statistically significant, it follows that a relationship between floods and previously extreme temperatures could be proved in the switching intensity model.

In case of the first follower model, the Wilcoxon rank-sum achieved a significant result. It means that the two distributions
are significantly different and the two processes are dependent.

In summary, it can be said that a dependency of floods and previously extreme temperatures could be determined using the switching intensity model and the first follower model.

### 5.5 Storms ⇒ Flood

Another possible reason for the occurrence of floods can be previous storms. Storms (ST) are often accompanied by heavy
rain, which of course leads to large amounts of water [9].

In terms of the switching intensity model, taking all countries into account, the results for estimating a Poisson process for flood would give the following estimation. Model validation via AIC shows that the third model, which includes both covariates, is the best because of the smallest AIC value.

According to the likelihood ratio test below, model 3 is in both cases the best model because both p-values are significant.
That is, including both covariates into the model improve the fit significantly.

Since the first model is statistically significant, it follows that a relationship between floods and previously storms could be proved in the switching intensity model.

In case of the first follower model, the Wilcoxon rank-sum achieved a significant result. It means that the two distributions are significantly different and the two processes are dependent.

---

[9]https://www.samhsa.gov/find-help/disaster-distress-helpline/disaster-types/floods





|  |  | Estimate | STD. Error | AIC |
|---|---|---|---|---|
| **MOD1** | Intercept | -3.102 | 0.066 | 1981.17 |
| (Dummy-ST) | Dummy | 1.818 | 0.245 |  |
|  |  |  |  |  |
| **MOD2** | Intercept | -4.497 | 0.184 | 1907.75 |
| (Time) | Time | 0.047 | 0.005 |  |
|  |  |  |  |  |
| **MOD3** | Intercept | -4.515 | 0.185 | 1883.67 |
|  | Time | 0.046 | 0.005 |  |
| (Dummy-ST+Time) | Dummy | 1.533 | 0.245 |  |

**Table 18.** Testing 3 models of FL

| Model 1: mod3 |  |  |  |  |  | Model 1: mod3 |  |  |  |  |  |
|---|---|---|---|---|---|---|---|---|---|---|---|
| Model 2: mod2 |  |  |  |  |  | Model 2: mod1 |  |  |  |  |  |
|  | #Df | LogLik | Df | Chisq | Pr(>Chisq) |  | #Df | LogLik | Df | Chisq | Pr(>Chisq) |
| 1 | 3 | -938.84 |  |  |  | 1 | 3 | -938.84 |  |  |  |
| 2 | 2 | -951.88 | -1 | 26.08 | 3.275e-07 *** | 2 | 2 | -988.58 | -1 | 99.499 | 2.2e-16 *** |

**Table 19.** Likelihood Ratio Test of FL depending on ST

In summary, it can be said that a dependency of both hazards could be determined using the switching intensity model and the first follower model.

## 6 Conclusion

Multi-hazard events can be devastating and there are indications that in such situations the exposed risk-bearers are affected more severe compared to single-hazard events as resources are drained at the same time, i.e. effecting their ability to respond. We presented some statistical modeling approaches to determine possible interrelationships of hazards and tested them for the specific case of the countries within the Danube Region. We especially focused on the question whether certain hazards are more likely to occur due to preceding hazards. The analysis presented here focused primarily on drought and wildfire hazards and possible relationships. The results of the analysis of the triggering effect showed for the *switching intensity model* that the consideration of previous droughts (up to 3 months) in addition to time significantly increases the intensity function according to the likelihood ratio test (p-value: 0.02). Furthermore, a significant dependency of the distribution of the first follower times



| Two-sample Wilcoxon rank-sum test |
| --- |
| T= 12457, p-value = 0.001036 |
| alternative hypothesis: two-sided |

**Table 20.** Testing of the dependnecy of the FL process on the ST process

and the inter-event times could be determined for the *first follower model* with the help of the Wilcoxon rank-sum test (p-value: 0.1). In summary, both models were able to show a significant dependency on wildfires with preceding droughts.

In a further step overviews of other possible relationships between different hazards were presented. Many of them could not be analyzed because of lack of data. Nevertheless, dependencies could be found for some relationships as the following
table shows

| relationship | significance for the switching intensity model | significance for the first follower model |
| --- | --- | --- |
| DR $\Rightarrow$ WF | yes | yes |
| ET $\Rightarrow$ DR | yes | no |
| ET $\Rightarrow$ WF | yes | yes |
| ET $\Rightarrow$ FL | yes | yes |
| ST $\Rightarrow$ FL | yes | yes |

**Table 21.** The investigated dependencies

Notice that the presented methodology aimed at revealing the relationship between the timing of hazard events of different type. Equally important is the question how the severity of a preceding event affects the severity of the triggered event. Recall that the damage curve models the relationship between severity of an event, exposure and vulnerability on one side and amount of damage on the other side. In some cases, the preceding event does not change severity of the triggered event, but increases
the vulnerabilty of the system. All this needs detailed investigation.

It must also be said that the small sample size of hazard observations can affect the reliability of this analysis. It is also clear that our analysis is restricted to the "Danube Region" and may not be valid for other geographical/geological/meterorological regions. The presented models and related tests in this paper, however, can be readily applied for other regions as well.

**Appendix A**

The following graphic shows a map of Europe with the countries to be considered, i.e. the Danube region.

The following pie chart represents the absolute and relative frequencies in percent for each disaster type.



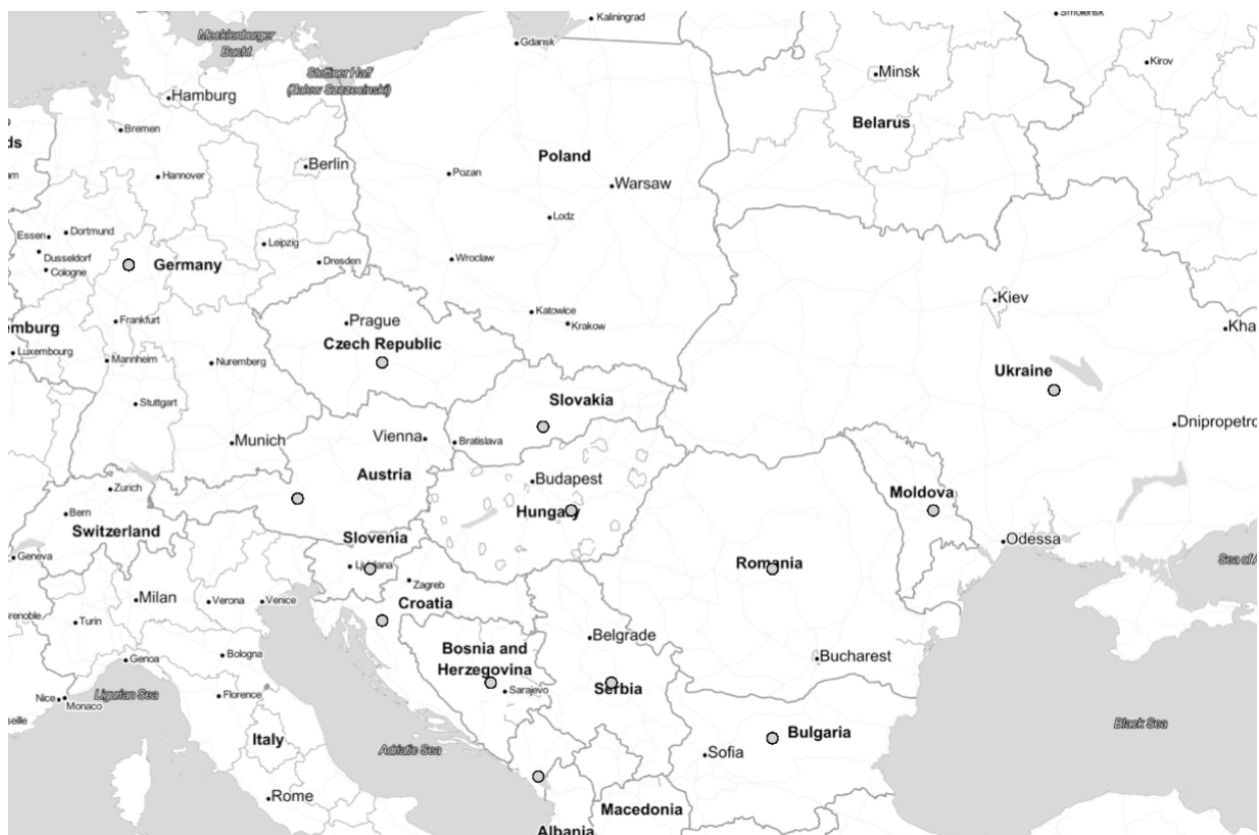

**Figure 9.** Map of Danube Region in Europe

*Author contributions.* Stefan Hochrainer-Stigler conceived and designed the research question and contributed material. Georg Pflug developed the mathematical and statistical models and Viktoria Kittler did the data handling, the implementation and the representation of the results. All three authors contributed in writing the paper.

*Competing interests.* There are no competing interests.

*Acknowledgements.* The work was done as a part of the HORIZON 2020 MYRIAD-EU Project, and the authors acknowledge the funding from the European Union's Horizon 2020 research and innovation programme call H2020-LC-CLA-2018-2019-2020 under grant agreement number 101003276.


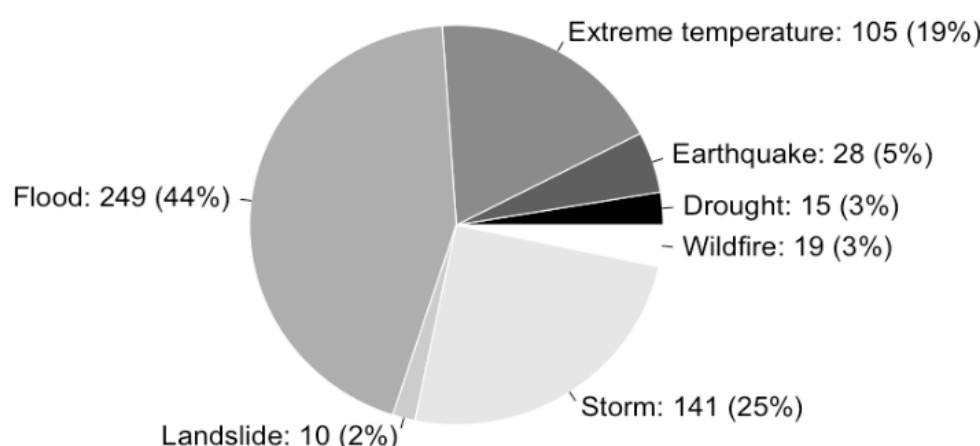

**Figure 10.** Pie chart of Disaster Types (n = 567)

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
