# Peer review of "Dependence Models for Multi-Hazard-Events"

_Natural Hazards and Earth System Sciences, 2023_

## Referee Comment (RC3)

Review of "Dependence Models for Multi-Hazard-Events"

I have been asked to act as a third reviewer of this manuscript.

This manuscript provides a mathematical approach to examining multi-hazards, which would be a welcome approach to the literature, but I do have concerns about the writing, formatting and the data used. Although I am critical below, I believe that with a MAJOR rewrite and thinking of the reader and how they read what is presented, some of the limitations and expressing those limitations for the data, and bringing in the wider literature in a number of places and themes, that this could be a very nice addition to the general literature on multi-hazards. However, this will require the authors to approach their writing in an appropriate way for the hazards community, and not 'assume' knowledge. I encourage the authors to do a major rewrite, perhaps working with others in the hazards community, so as to bring these statistical approaches into a working framework for those who might not be as mathematically inclined.

My comments, not in order of importance.

1. **Abstract**. This is more of an introduction, not a summary of the paper. This could be much stronger.
2. **Introduction**.
   a. This provides a good motivation at the beginning, but then seems to all of a sudden go into the main part of the manuscript core ideas. I was a bit lost at the abrupt change and flow.
   b. What does it matter if these are Gauss-Krüger coordinates (why bring these in?) [Line 28].
   c. Although I am familiar with Gill and Malamud (2014) some readers will not be. So they will not understand (line 34, what it means to have a 21 x 21 table. 21 'what'? So you say 21 types of hazards, but this is unclear that the same 21 hazards are being considered. Ah, I see now, you are discussing your Table 1 (without referring to it) and trying to illustrate it but you never actually tell us 'See Table 1'. Argh! This was difficult to make this interpretation that is what we are supposed to look at as the Table 1 is just 5 x 5.
   d. In Gill and Malamud (2014) which I've read, you have equated their degree of dependency (which you state goes from triggering as highest degree of dependency then to lower degree of dendency as increased probability) with spatial-temporal overlap. This is not correct. They are not the same thing. I'm also unclear how your 0 to 3 relates to their spatial-temporal overlap factor. Lines 38 to 50 you should revisit your argument. Note that this discussion you have does not influence your methodology, so is minor.

e. Please provide at the end of Section 1 how you will organize the rest of the manuscript. This is standard in most papers.

3. **Methodology**

   a. There is a large literature on the recurrence time between discrete natural hazard events (e.g., earthquakes, floods), and inter-event occurrence times (and their statistical distributions). Do you want perhaps to build on some of this literature? It seems a rather a type of hubris for you to only cite yourself when an extensive literature exists.

   b. I found that despite the lack of understanding of others who have worked in the literature on hazard events and inter-event occurrence time distributions, the rest of Section 2 Methodology well written and interesting. In almost all cases though I would have liked to have better understood who else has worked with these methodologies in the hazard communities—it feels rather exclusionary to what many in the geostatistical community have done, but is still a worthwhile methodology.

4. **Case Study: Danube**

   a. Please tell us a bit more about the Danube Region in terms of its geophysical/hydrological/weather characteristics so as to give us a background.

   b. Please put the map into the manuscript (not as an appendix).

   c. Data: I am fundamentally not convinced by your use of EM-DAT for this purpose. At the very least you need to understand its limitations before doing an analysis like this, and discuss them. The events that EM-DAT records are highly biased towards those with x fatalities, y economic damage, or a triggering of disaster risk platforms. This will highly influence the data set you use. I'm not saying your methodology is wrong or the results necessarily wrong, but you really do need to acknowledge these rather severe limitations.

   d. Tell us the period of time over which you are using the EMDAT data set and whether these data have changed over time (there is often an undersampling earlier on, and these might change if human population densities have changed).

   e. Put your pie chart (which I don't recommend for a scientific audience) in the text as a figure, not in the appendix.

   f. I like the idea of your using timing and location in Table 3, but question what happens for delayed periods (e.g., a severe landslide might occur months after a fire).

5. **Analysis and Results**

   a. I was finding this a bit more difficult to follow some of the flow of the argument.

b. The graphs were helpful (first time looking at the data) although a tad on the basic side of things in terms of visualisation.

c. I'm not sure of the point of Figure 2b; isn't 2a enough? If not, state why it is there. Is this for some sor tof 'intensity' measure?

d. For droughts, they have a period of time (sometimes weeks, sometimes months or years). Is this represented in your analysis?

e. I find it difficult with my knowledge of droughts to equate 'frequency' with intensity. There is so much more that gets into a drought intensity. Perhaps you could argue that only those droughts over a certain threshold are brought into EM-DAT, so that by counting the peaks of threshold this is an alternative measure of intensity?

f. A find it a bit odd to think about droughts 'triggering' wildfires, in the sense used by Gill and Malamud (2014). Droughts can trigger but often increase the probability of the conditions changing such that a wildfire might occur.

6. **Formatting/grammar.**

a. Tables. These are poorly brought into the argument, and make this a very difficult manuscript to read.

   i. There is an odd placement of some tables in the middle of paragraphs. Perhaps this was an oddity of the software package used (looks ike LaTeX).

   ii. Tables 1, 3, 7 are the ONLY tables out of 21 tables actually mentioned in the text by number. For the others, one had to 'guess' which ones are being referred to (it might say 'in the following table', but then there was no table!). This made for really difficult reading.

   iii. Table 1. Great, it is mentioned, but the table comes two pages before it was mentioned.

b. Some odd characters appearing here or there (e.g., "@" on line 74).

c. Please avoid phrases like "It is quite obvious". If it is so obvious, then don't state it. And, this is 'not' obvious (line 33).

d. Avoid having 'e.g.' at beginning of a list and 'etc.' at the end. One or the other please.

e. Repeated words like "Gill and Malamud Gill and Malamud" line 39.

f. Strange use of capitals in places (e.g., Table 1, why is it "Example"?).

g. Table captions are not very complete or self-standing, so that the table could be removed from the manuscript without having to go to the text. This is generally considered poor practice. For example, I would have no idea what table 1 is without reading the text. The EQ, LA, FL are not defined, nor the meaning of 0, 1, 2, 3.

h. Three " on lines 46-47, so unclear what the quote is.

      i.    Please avoid footnotes. This is considered very unusual for writing in these types of journals.

      j.    Font size on some figures getting to VERy small (e.g., Figure 7).

      k.    March 8h (line 236)

7. **Referencing**.

    a.    There seems a paucity of references in this work and general understanding that you are building on what others have done, rather than reinventing the wheel. What you are doing is interesting, but please do recognize the general community and what has been done.

    b.    Some statements (please check throughout) need referencing. So for example lines 33 and 34, your 'obvious' statement.

    c.    There is a literature on looking both statistically and deterministically at how one hazard can trigger another. There is little mention of this literature.